# Parkinsonism Risk Factors in Salt Lake City, Utah: A Community-Based Study

**DOI:** 10.3390/brainsci9030071

**Published:** 2019-03-23

**Authors:** David Shprecher, Nan Zhang, Matthew Halverson, Rodolfo Savica

**Affiliations:** 1Banner Sun Health Research Institute, 10515 W. Santa Fe Drive, Sun City, AZ 85351, USA; 2Department of Neurology, University of Utah, 175 North Medical Drive East, Salt Lake City, UT 84132, USA; 3Department of Neurology, University of Arizona, Phoenix, AZ 85724, USA; 4Department of Biostatistics, Mayo Clinic, Scottsdale, AZ 85259, USA; Zhang.Nan@mayo.edu; 5Huntsman Cancer Institute, University of Utah, Salt Lake City, UT 84112, USA; matt.halverson@hci.utah.edu; 6Department of Neurology, Mayo Clinic, Rochester, MN 55905, USA; Savica.Rodolfo@mayo.edu

**Keywords:** parkinsonism, Parkinson’s disease, anosmia, hyposmia, psychosis, dementia with Lewy bodies, REM sleep behavior disorder, constipation, mild cognitive impairment

## Abstract

Background: The prevalence of dream enactment behavior and other risk factors for a parkinsonian disorder is not well documented. Methods: A survey on prevalence of parkinsonism risk factors was designed using two validated instruments (REM behavior disorder single item question, bowel movement frequency for constipation) and three exploratory instruments (for hallucinations, cognitive and olfactory complaints.) It was sent by mail and email to patients aged 50 and over at two University of Utah community clinics in Salt Lake City. A total of 7888 unique patients were sent the survey, and 1607 responses were recorded (response rate 20%). Those whose age was missing (*n* = 117) or less than 50 years (*n* = 10) were excluded from the analysis. Results: Of the 1406 without personal diagnosis of neurodegenerative disease 62.7% were female, and median age was 63. Family history (FH) of Parkinson’s disease was endorsed by 9%, constipation (defined as a bowel movement less than once per day) by 19%, mild cognitive complaints (MCI) 15.8%, dream enactment 13.7%, subjective hyposmia or anosmia 18.2%, and at least one potential psychotic symptom in 37.6%. Multivariable logistic regression showed male gender, mild cognitive complaints, hearing voices, and at least one potentially psychotic symptom to be significantly associated with dream enactment. Conclusions: This survey shows that dream enactment, a strong predictor of risk for synucleinopathy, is relatively common in the older population; because such individuals rarely come to medical attention of a sleep clinic, such survey research may be useful to identify and recruit at-risk individuals for trials aimed at preventing neurodegenerative disease.

## 1. Introduction

Dementia with Lewy bodies (DLB) and Parkinson’s disease (PD) dementia, collectively known as Lewy body dementia (LBD), are the leading causes of neurodegenerative parkinsonism and the second-leading cause of age-associated neurodegenerative dementia [1,2]. With a growing geriatric population, there is an urgent need to develop early detection and prevention strategies for LBD. Interpretation of neuroprotection trials in manifest parkinsonian disorders may be confounded by direct effects of the treatment on clinical measures of disease progression, and hindered by the fact that a majority of dopaminergic neurons have already been lost by the time of diagnosis. Performing neuroprotection trials in a premanifest population at imminent disease risk could circumvent these problems.

Robust clinical and epidemiological evidence has shown idiopathic REM sleep behavior disorder (iRBD) to be the strongest known clinical predictor of neurodegenerative alpha-synucleinopathy (DLB, PD, or multiple system atrophy), particularly LBD [3,4,5]. This parasomnia, characterized by dream enactment behavior with loss of REM-related muscle atonia, has been associated with an 80–91% lifetime risk of an alpha-synucleinopathy (PD, DLB, or multiple system atrophy) [6,7]. Early dementia appears to be the rule rather than the exception, and its risk may be further predicted with olfactory and color vision testing. Of the 21 iRBD subjects who developed alpha-synucleinopathy, the 17 who developed dementia in the first 2 years of illness had olfactory and color vision deficits at baseline [8].

In addition to iRBD, clinical symptoms of anosmia, color vision impairment, constipation, mild cognitive impairment (MCI), and psychosis are common, often prodromal, features of PD and related disorders [8,9,10,11,12,13,14,15]. We conducted a survey on risk factors for neurodegenerative disorders in our population aged 50 and over, in order to ascertain their frequency and guide design and recruitment strategies for studies of premanifest disease.

## 2. Materials and Methods

The University of Utah Institutional Review Board (IRB) approved the conduct of this study with a waiver of written informed consent using a survey cover letter. Only University employees trained and delegated by the principal investigator (DS) had access to date of birth information (collected in order to calculate age at time of survey completion, and to monitor for duplicate responses.).

### 2.1. Survey Content

A cover letter explaining the goal of the survey (signed by the University of Utah Community Clinics’ Medical Director) was included. The investigators (DS and RS) designed the survey. Dream enactment was assessed using a validated single item question for probable RBD, [16] constipation with a bowel movement frequency question adapted from a validated instrument, [17] and potential psychotic symptoms with questions modeled from a validated instrument for PD psychosis, MCI, and anosmia with novel questions. In addition to demographic information (gender, date of birth, education, and date of survey completion) the following symptom-specific questions were asked:(1)Sleep Symptoms(a) RBD single question: ”Have you ever been told, or suspected yourself, that you seem to act out your dreams (for example, punching; flailing your arms; making running movements; shouting out loud; knocking things over; jumping out of bed)?”(b) History of sleep walking behaviors(2)MCI: ”Do you or any of your family/friends suspect that you have more problems with thinking than most people your age (forgetting events or conversations, missing appointments, trouble coming up with names or words, getting lost in familiar places)?”(3)Anosmia: “How would you rate your sense of smell (choose one)? Normal, not as good as other people my age, very poor, absent, never had much of a sense of smell.”(4)Psychosis: yes or no questions to each, asked for the past year “Have you ever seen vague shapes, lights, or spots that weren’t really there”; “Have you ever seen objects, people, animals, or scenery that weren’t really there?”; “Have you ever heard voices, music, or other complex sounds that weren’t really there?”; “Have you ever felt like a person or animal was with you when they were not?”; “Have you ever felt a person or animal touching you when they were not?”; “Have you ever had paranoid thoughts that others would not agree with?”(5)Constipation: “How often do you have a bowel movement? More than three times a day, two or three times a day, once a day, about every other day, More than once a day, once a day, about twice a week, about once a week, less than once a week.”(6)Neurological history (diagnosis of Parkinson’s, Alzheimer’s disease, or a related condition)(7)Family history of neurodegenerative disease (by type: PD, dementia, and/or atypical parkinsonism).

In order to facilitate recruitment for future RBD cohort or neuroprotection trials, respondents could provide consent for future contact by entering full name, telephone number, and/or email address. A total of 172 participants fitting these criteria provided consent to be contacted for future research.

### 2.2. Recruitment and Data Collection

Two primary care clinics (Madsen and Sugarhouse) in close proximity to the University of Utah, Salt Lake City were chosen for this study. Cards were placed in each clinic waiting room explaining that patients may receive a research survey at their email or home address of record. Cards included instructions on how to opt out from receiving surveys. From June 30th, 2014 to April 30th, 2015 this survey was sent by mail (with self-return envelopes) to all patients 50, and was overseen at these clinics. For those with an email address available, a link to the web-based survey was also sent. Online respondents’ data was imported directly to a RedCap database. Mail respondents’ data was entered by study personnel using the RedCap survey page. Returned surveys were received and imported through December 2015.

### 2.3. Data Analysis

As surveys were sent more than once to patients with more than one follow up during the survey time frame, duplicate responses could be received. Responses were deemed to be duplicates if they showed the same date of birth, gender, education level, address and name (if provided), and responses for risk factor questions. In these cases, only data from the first survey received was analyzed. Frequencies of responses were calculated and stratified by presence or absence of a neurological diagnosis that could cause RBD. Two sample *t*-test or Chi-square tests were used when applicable to compare demographics and clinical characteristics between groups with and without neurological diagnoses. Demographics, family history of neurodegenerative disease, and potential risk factors were each examined with univariate logistic regression for association with the presence of probable RBD (pRBD). Variables with *p* < 0.2 were chosen into the multiple logistic regression and backward selection with 0.05 as the cut-off was used to choose the variables in the final multivariable model. These univariate and multivariable analyses were completed for the entire survey population and separately for the idiopathic pRBD subjects (who did not report diagnosis of a potential secondary cause of RBD.)

## 3. Results

A total of 7888 surveys were sent. Of 1770 received, 163 were deemed duplicates, for a total of 1607 respondents (response rate 20.3%.) Of these, 74/1480 (5%) reported diagnosis of PD, Alzheimer’s disease (AD), or another neurological disease (brain cancer, demyelinating disease, stroke, traumatic brain injury, or vascular dementia) that could potentially cause secondary RBD (33/1480 (2.2%) PD, 5/1480 (0.3%) AD, 9/1480 (0.6%) dementia, 26/1480 (1.8%) other). Demographics and risk factors reported for the entire population, stratified by presence or absence of a neurological diagnosis, are detailed in Table 1. All risk factor variables except family history were significantly more common (based upon a cutoff of *p* < 0.05) among individuals with a neurological diagnosis, who were significantly more likely to report two or more psychotic symptoms (though significance was not met for number reporting at least one psychotic symptom.) A total of 219/1480 (14.8%) of all respondents, and 13.77% of those without a neurological diagnosis, reported pRBD (based on history of dream enactment). Univariate analysis for the entire population showed potential association between RBD and male gender, college degree, mild cognitive complaints, first degree relative with neurodegenerative disease (other than PD or dementia), olfactory complaints, and potential psychotic symptoms. Male gender, first degree relative, mild cognitive complaints, hearing voices, seeing objects, and at least one potential psychotic symptom retained significance in multivariable analysis (Table 2). For those with idiopathic pRBD (respondents who endorsed dream enactment but not history of potentially secondary causes such as PD), the same associations were seen in univariate analysis and in multivariable analysis significance was retained for male gender, mild cognitive complaints, hearing voices, and at least one potentially psychotic symptom (Table 3).

## 4. Discussion

We found that self-reported, potentially prodromal, symptoms of PD or related disorders to be common in the general population. The prevalence of psychotic symptoms may at first glance seem extraordinarily high. However, seeing shapes (endorsed by 27% overall) could potentially be related to age-related visual problems, whereas prevalence for seeing objects (6.6%) or hearing voices (8.5%) is comparable to reported prevalence for hallucinations in any sensory modality (9%) within a French population aged 60 and older [18]. Impaired olfaction and constipation were each reported in about 20% of the population. However, these risk factors may not be sufficient to identify individuals at imminent risk for parkinsonian disorders. Even when combined with abnormal substantia nigra (SN) echogenicity, anosmia only showed a positive predictive value of 6.1% for phenoconversion after three years [19]. When combined with abnormal imaging biomarkers (SN echogenicity or Dopamine transporter SPECT), 30% of individuals with PSG confirmed diagnosis of RBD developed a parkinsonian disorder within 2.5 years of follow-up [20]. Our rate of pRBD, 15.1% overall, was higher than a recent door to door survey of a Chinese community where the rate was 4.9% (but a much broader age range, 20 to 99, was assessed) [21]. Further research using the RBD single item question is needed to validate our findings in a population with similar demographics to ours. The RBD single question item employed in our survey has shown 93.8 % sensitivity and 87.2% specificity in identifying individuals with pRBD who would likely have RBD confirmed if they underwent polysomnogram (PSG.) [16]. However, these were measured in a sleep clinic population with a high prevalence of neurodegenerative disease. When recruited from a community based sample in Montreal (with similar demographics to our own), 17.1% of subjects screened for secondary causes (antidepressant use) or RBD mimics (sleep apnea, and restless legs), and referred (where appropriate) for PSG were confirmed to have definite RBD [22]. Thus, the most conservative estimate for frequency in our general population, would be 2.6% for overall RBD and 2.3% for iRBD. This is comparable to a previous study using PSG confirmation in an elderly Korean population, where prevalence was 2.1% for RBD, 1.15% for idiopathic RBD, and 4.95% for subclinical RBD (REM sleep without atonia, in absence of notable dream enactment behavior) [23].

This study has important limitations. Due to limited staffing and funding, we were not able to send reminders to individual participants, and the overall response rate was modest. This could have led to respondent bias, with individuals who have personal or family history of neurodegenerative disease more likely to participate. However, the prevalence of neurodegenerative and related disorders was 5%, and family history thereof was not associated with pRBD in our sample. We cannot exclude other forms of respondent bias in favor of (or against) response among those with symptoms of interest. The questions about potential psychotic symptoms were adapted from a validated instrument developed for PD [20]. However, this instrument is designed for in-person interview and has not yet been validated for written surveys of the general population.

## 5. Conclusions

When we combined all predictors, male gender, mild cognitive complaints, and psychotic symptoms were associated with RBD. To our knowledge, psychotic symptoms have not been routinely assessed in prospective research on predictors of phenoconversion in iRBD. Our findings suggest a screening instrument for psychotic symptoms would be worth including in such research. Though subjective hyposmia did not retain an association with iRBD in multivariate analysis, objectively measured hyposmia has been strongly associated with risk of phenoconversion in prospective iRBD studies. Taken together, our data and the previously published research [19,24,25] suggest that a broad, multiple-step screening and recruitment strategy will be necessary in order to recruit for prevention trials, once promising therapeutic strategies have been chosen.

## Figures and Tables

**Table 1 brainsci-09-00071-t001:** Demographics and Clinical Characteristics for Overall Population Separated by Neurodegenerative Disease Status.

	No Conditions (*n* = 1406)	Conditions * (*n* = 74)	*p*-Value **
Age			0.0038
Mean, SD	64.7 (9.9)	68.1 (11.5)	
Median (Range)	63.0 (50.0, 100.0)	66.5 (50.0, 100.0)	
Female, *n* (%)	853 (62.7%)	31 (41.9%)	0.0003
Education Level, *n* (%)			0.4337
Less than high school	37 (2.6%)	1 (1.4%)	
High school/trade school certificate	303 (21.6%)	19 (25.7%)	
Some college or an associate degree	143 (10.2%)	11 (14.9%)	
Bachelor’s degree	403 (28.8%)	22 (29.7%)	
Graduate degree	514 (36.7%)	21 (28.4%)	
Living with others, *n* (%)	1011 (72.2%)	62 (83.8%)	0.0287
Sleeping with a bed partner, *n* (%)	753 (54.0%)	42 (57.5%)	0.5523
First degree relatives had PD, *n* (%)	124 (9.0%)	11 (15.1%)	0.081
First degree relatives had dementia, *n* (%)	392 (28.5%)	19 (26.0%)	0.6494
First degree relative had nds,	120 (8.9%)	12 (16.9%)	0.0232
Ever acted out of dreams, *n* (%)	191 (13.7%)	28 (37.8%)	<0.0001
Sleep walking behaviors, *n* (%)	104 (7.5%)	6 (8.1%)	0.8422
Problems with thinking,	219 (15.8%)	39 (52.7%)	<0.0001
Subjective olfaction, *n* (%)	252 (18.2%)	30 (40.5%)	<0.0001
Constipation, *n* (%)	264 (19.0%)	26 (35.6%)	0.0005
Seen shapes, *n* (%)	376 (27.1%)	29 (39.7%)	0.0185
Seen objects, *n* (%)	92 (6.6%)	18 (24.7%)	<0.0001
Heard voices, *n* (%)	118 (8.5%)	14 (19.4%)	0.0016
Sensed presence, *n* (%)	115 (8.3%)	16 (22.5%)	<0.0001
Tactile hallucination, *n* (%)	71 (5.1%)	11 (15.1%)	0.0003
Paranoid thoughts, *n* (%)	64 (4.6%)	10 (13.5%)	0.0007
Told you you were acting paranoid, *n* (%)	79 (5.8%)	11 (14.9%)	0.0016
At least one psychotic symptom, *n* (%)			<0.0001
No psychotic symptoms	872 (62.4%)	41 (55.4%)	
Only 1 psychotic symptom	334 (23.9%)	9 (12.2%)	
2+ psychotic symptom	192 (13.7%)	24 (32.4%)	

*: Conditions were defined as dementia/neurodegenerative disease, brain cancer, TBI, demyelinating diseases, stroke; **: Two sample *t*-test or Chi-square test was used when applicable; nds = neurodegenerative disease.

**Table 2 brainsci-09-00071-t002:** Multiple logistic regression results among all subjects.

Variable	Effect	OR (95% CI)	*p* Value
Gender	Male vs. Female	1.80 (1.30, 2.50)	0.0004
first degree relative had another nds	Yes vs. No	1.69 (1.07, 2.69)	0.0254
Problems with thinking	Yes vs. No	2.35 (1.63, 3.40)	<0.0001
Seen objects	Yes vs. No	1.80 (1.06, 3.07)	0.0306
Heard voices	Yes vs. No	1.81 (1.10, 2.97)	0.0194
At least one psychotic symptom	Yes vs. No	2.05 (1.41, 2.96)	0.0001

nds = neurodegenerative disease.

**Table 3 brainsci-09-00071-t003:** Multiple logistic regression results among p-iRBD cases.

Variable	Effect	OR (95% CI)	*p* Value
Gender	Male vs. Female	1.82 (1.29, 2.57)	0.0007
Problems with thinking	Yes vs. No	2.42 (1.63, 3.61)	<0.0001
Heard voices	Yes vs. No	1.81 (1.09, 3.00)	0.0223
At least one psychotic symptom	Yes vs. No	2.44 (1.67, 3.54)	<0.0001

p-iRBD = probable idiopathic REM sleep behavior disorder.

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
