# Peer review of "Parkinsonism Risk Factors in Salt Lake City, Utah: A Community-Based Study"

_brainsci, 2019, doi:10.3390/brainsci9030071_

Reviewer 1 Report

brainsci-457381-peer-review-v1
Parkinsonism risk factors in Salt Lake City, Utah: a Community-Based Study David Shprecher et al
This interesting manuscript reports the compendium of a mail-in or online survey of patients concerning Parkinsonism risk factors. The results are potentially interesting – the data seem to argue that dream enactment (idiopathic REM sleep behavior disorder; RBD) is a strong clinical predictor of alpha-synucleinopathy (neurodegeneration) – but the data strictly deal only with the questionnaire, not with underlying (potential) neurodegenerative processes.
The questions in the questionnaire seem perfectly reasonable. The paper needs to address two seemingly disparate findings: the level of potential psychotic symptoms (38% seems very high) and the response rate to the questionnaire (20%) which seems to this reviewer to be very low, as acknowledged later in the paper. The obvious interaction of these numbers is that the low response rate could lead to the high psychotic symptoms rate by a reverse selection – perhaps family members complete the form for the potentially psychotic patients? Anyhow, the two numbers raise linked questions about this study which the authors need to address. The rate (15%) of respondents reporting RBD seems very high and needs to be addressed with literature citations and discussion.
There are additional places in the manuscript where the authors miss a golden opportunity to explain/discuss their data. For example, lines 168-170, “the response rate was modest, and likely higher….” So does this lead to a serious skewing of the results? Maybe, maybe not – this is for the authors to address and for the readers and reviewers to judge – it should not be left hanging.
If one wanted to be a stickler for clarity, the Funding statement is confusing and ambiguous. There is no “DRS” on the author list – this could conceivably refer to the first author (DS) or the last author (RS) depending on whether the last author goes by the middle name and drops a lead ‘D’.  This needs to be clarified.

Author Response

p.p1 {margin: 0.0px 0.0px 0.0px 0.0px; font: 12.0px 'Helvetica Neue'; color: #454545} p.p2 {margin: 0.0px 0.0px 0.0px 0.0px; font: 12.0px 'Helvetica Neue'; color: #454545; min-height: 14.0px}

This interesting manuscript reports the compendium of a mail-in or online survey of patients concerning Parkinsonism risk factors. The results are potentially interesting – the data seem to argue that dream enactment (idiopathic REM sleep behavior disorder; RBD) is a strong clinical predictor of alpha-synucleinopathy (neurodegeneration) – but the data strictly deal only with the questionnaire, not with underlying (potential) neurodegenerative processes. We have (to explain the rationale for our survey on RBD and other risk factors) cited relevant literature in the background section showing that idiopathic RBD is a strong predictor of synucleinopathy. The data  in the results section deal strictly with findings of our  survey.
The questions in the questionnaire seem perfectly reasonable. The paper needs to address two seemingly disparate findings: the level of potential psychotic symptoms (38% seems very high) and the response rate to the questionnaire (20%) which seems to this reviewer to be very low, as acknowledged later in the paper. The obvious interaction of these numbers is that the low response rate could lead to the high psychotic symptoms rate by a reverse selection – perhaps family members complete the form for the potentially psychotic patients? Anyhow, the two numbers raise linked questions about this study which the authors need to address.

In our discussion section, we have pointed out that “seeing shapes (endorsed by 27% overall) could potentially be related to age-related visual problems, whereas prevalence for seeing objects (6.6%) or hearing voices (8.5%) is comparable to reported prevalence for hallucinations in any sensory modality (9%) within a French population aged 60 and older.”  We have added a statement to our limitations section that “We cannot exclude other forms of respondent bias in favor of (or against) response among those with symptoms of interest.

The rate (15%) of respondents reporting RBD seems very high and needs to be addressed with literature citations and discussion.  We have added this important point to the discussion.

There are additional places in the manuscript where the authors miss a golden opportunity to explain/discuss their data. For example, lines 168-170, “the response rate was modest, and likely higher….” So does this lead to a serious skewing of the results? Maybe, maybe not – this is for the authors to address and for the readers and reviewers to judge – it should not be left hanging.  We have expanded this discussion of the limitations as follows, “This could have led to respondent bias, with individuals who have personal or family history of neurodegenerative disease more likely to participate.  However; prevalence of neurodegenerative and related disorders was 5% and family history thereof was not associated with pRBD in our sample.  We cannot exclude other forms of respondent bias in favor of (or against) response among those with symptoms of interest.“ 
If one wanted to be a stickler for clarity, the Funding statement is confusing and ambiguous. There is no “DRS” on the author list – this could conceivably refer to the first author (DS) or the last author (RS) depending on whether the last author goes by the middle name and drops a lead ‘D’.  This needs to be clarified. This has been corrected (middle initial removed).

Reviewer 2 Report

The authors tackle the  topical issue of identifying an at risk population for potential disease modifying treatment trials in neurodegenerative disease. They use a survey to assess the frequency of known risk factors for alpha-synucleinopathies. Overall this work is interesting and provides a general approach to identifying an at risk cohort for PD. It is not exceptionally novel, given the work of the PREDICT-PD study (see below), but still has merit in providing prevalence of at-risk features  in a large community-based population.

I would like to know in more detail how the questionnaires were constructed - were they written by the study team, or did they come from another validated source? This is discussed to some extent in the discussion, but would be better placed in the methods.

Were reminders sent to those who did not reply to the first invitation?

I would suggest the authors consider how their work complements that of the PREDICT-PD study (eg https://jnnp.bmj.com/content/85/1/31.short and  https://www.ncbi.nlm.nih.gov/pubmed/23071076) - at least some of this work needs to be cited in the introduction.

The statistical methods are robust and the results are well presented and clear.

The discussion is appropriately cautious in interpreting the results.

Author Response

p.p1 {margin: 0.0px 0.0px 0.0px 0.0px; font: 12.0px 'Helvetica Neue'; color: #454545} p.p2 {margin: 0.0px 0.0px 0.0px 0.0px; font: 12.0px 'Helvetica Neue'; color: #454545; min-height: 14.0px}

The authors tackle the  topical issue of identifying an at risk population for potential disease modifying treatment trials in neurodegenerative disease. They use a survey to assess the frequency of known risk factors for alpha-synucleinopathies. Overall this work is interesting and provides a general approach to identifying an at risk cohort for PD. It is not exceptionally novel, given the work of the PREDICT-PD study (see below), but still has merit in providing prevalence of at-risk features  in a large community-based population.  I would like to know in more detail how the questionnaires were constructed - were they written by the study team, or did they come from another validated source? This is discussed to some extent in the discussion, but would be better placed in the methods.

We have added these details to the methods section. 

Were reminders sent to those who did not reply to the first invitation?

Thank you for this important question; we have added a clarification in the discussion section, Due to limited staffing and funding, we were not able to send reminders to individual participants and overall response rate was modest.”

I would suggest the authors consider how their work complements that of the PREDICT-PD study (eg https://jnnp.bmj.com/content/85/1/31.short and  https://www.ncbi.nlm.nih.gov/pubmed/23071076) - at least some of this work needs to be cited in the introduction.

Thanks for these additional citation recommendations. We have added citation the meta-analysis by Noyce et al 2012 to our introduction (providing further support for inclusion of family history and constipation in this survey).  The PREDICT-PD study, as well as the PRIPS study (Berg et al) are  that (while they include survey components) are longitudinal studies designed to assess utility of specific assessments in predicting risk of a parkinsonian syndrome.  We have added appropriate citation of these (and similarly relevant work) in the concluding sentence of our discussion.   

The statistical methods are robust and the results are well presented and clear.
Thank you.

The discussion is appropriately cautious in interpreting the results.  Thank you.

Round  2

Reviewer 1 Report

the authors have addressed all the concerns